

# The SP19 Chronology for the South Pole Ice Core - Part 1:
## Volcanic matching and annual-layer counting
Dominic A. Winski[1,2], Tyler J. Fudge[3], David G. Ferris[4], Erich C. Osterberg[4], John M.
Fegyveresi[5], Jihong Cole-Dai[6], Zayta Thundercloud[4], Thomas S. Cox[7], Karl J. Kreutz[1,2],
Nikolas Ortman[4], Christo Buizert[8], Jenna Epifanio[8], Edward J. Brook[8], Ross Beaudette[9],
Jeff Severinghaus[9], Todd Sowers[10], Eric J. Steig[3], Emma C. Kahle[3], Tyler R. Jones[11],
Valerie Morris[11], Murat Aydin[12], Melinda R. Nicewonger[12], Kimberly A. Casey[13,14],
Richard B. Alley[10], Edwin D. Waddington[3], Nels A. Iverson[15], Ryan C. Bay[16], Joseph M.
Souney[17]
[1]*School of Earth and Climate Sciences, University of Maine, Orono, Maine, USA*
[2]*Climate Change Institute, University of Maine, Orono, Maine, USA*
[3]*Department of Earth and Space Sciences, University of Washington, Seattle,*
*Washington, USA*
[4]*Department of Earth Sciences, Dartmouth College, Hanover, New Hampshire, USA*
[5]*U.S. Army Corps of Engineers Cold Regions Research and Engineering Laboratory,*
*Hanover, New Hampshire, USA*
[6]*Department of Chemistry and Biochemistry, South Dakota State University, Brookings,*
*South Dakota, USA*
[7]*Physical Science Department, Butte College, Oroville, California, USA*
[8]*College of Earth, Ocean and Atmospheric Sciences, Oregon State University, Corvallis,*
*Oregon, USA*
[9]*Scripps Institution of Oceanography, UC San Diego, La Jolla, California, USA*
[10]*Department of Geosciences and Earth and Environmental Systems Institute,*
*Pennsylvania State University, University Park, Pennsylvania, USA*
[11]*Institute of Arctic and Alpine Research, University of Colorado, Boulder, Colorado,*
*USA*
[12]*Department of Earth System Science, UC Irvine, Irvine, California, USA*
[13]*Earth Sciences Division, NASA Goddard Space Flight Center, Greenbelt, MD, USA*
[14]**now at National Land Imaging Program, U.S. Geological Survey, Reston, VA, USA*
[15]*New Mexico Institute of Mining and Technology, New Mexico Bureau of Geology and*
*Mineral Resources, Socorro, New Mexico, USA*
[16]*Physics Department, University of California, Berkeley, California, USA*
[17] *Institute for the Study of Earth, Oceans and Space, University of New Hampshire,*
*Durham, New Hampshire, USA*
*Correspondence to:* Dominic Winski (dominic.winski@maine.edu)





## Abstract


The South Pole Ice Core (SPICEcore) was drilled in 2014-2016 to provide a
detailed multi-proxy archive of paleoclimate conditions in East Antarctica during the
Holocene and late Pleistocene. Interpretation of these records requires an accurate depth-
age relationship. Here, we present the SP19 timescale for the age of the ice of SPICEcore.
SP19 is synchronized to the WD2014 chronology from the West Antarctic Ice Sheet
Divide (WAIS Divide) ice core using stratigraphic matching of 251 volcanic events.
These events indicate an age of 54,302 +/- 519 years BP (before the year 1950) at the
bottom of SPICEcore. Annual layers identified in sodium and magnesium ions to 11,341
BP were used to interpolate between stratigraphic volcanic tie points, yielding an
annually-resolved chronology through the Holocene. Estimated timescale uncertainty
during the Holocene is less than 18 years relative to WD2014, with the exception of the
interval between 1800 to 3100 BP when uncertainty estimates reach +/- 25 years due to
widely spaced volcanic tie points. Prior to the Holocene, uncertainties remain within 124
years relative to WD2014. Results show an average Holocene accumulation rate of 7.4
cm/yr (water equivalent). The time variability of accumulation rate is consistent with
expectations for steady-state ice flow through the modern spatial pattern of accumulation
rate. Time variations in nitrate concentration, nitrate seasonal amplitude, and $\delta^{15}N$ of $N_2$
in turn are as expected for the accumulation-rate variations. The highly variable yet well-
constrained Holocene accumulation history at the site can help improve scientific
understanding of deposition-sensitive climate proxies such as $\delta^{15}N$ of $N_2$ and photolyzed
chemical compounds.

## 1. Introduction

Polar ice core records provide rich archives of paleoclimate information that have
been used to advance understanding of the climate system. One of the great strengths of
ice cores is the tightly constrained dating that permits interpretation of abrupt events and
comparisons of phasing among records. Therefore, a critical phase in the development of
any ice core record is the rigorous establishment of a depth-age relationship.
Several techniques are available to assign ages to each specific depth in an ice
core. These include annual layer identification of chemical (e.g. Sigl et al. 2016;
Andersen et al. 2006; Winstrup et al. 2012) and physical (e.g. Hogan and Gow 1997;
Alley et al. 1997) ice properties, identification of stratigraphic horizons as relative age
markers (e.g. Sigl et al. 2013; Bazin et al. 2013; Veres et al. 2013) and glaciological flow
modeling (e.g. Parrenin et al. 2004). To establish a depth-age relationship for the South
Pole Ice Core (hereafter SPICEcore), we use a combination of 1) annual layer counting of
glaciochemical tracers and 2) stratigraphic matching of volcanic horizons to the West
Antarctic Ice Sheet (WAIS) Divide ice core timescale "WD2014" (Sigl et al. 2016,
Buizert et al. 2015).
SPICEcore was drilled in 2014-2016 for the purpose of establishing proxy
reconstructions of temperature, accumulation, atmospheric circulation and composition,
and other earth system processes for the last 40,000 years (Casey et al. 2014). The
SPICEcore record is the only ice core south of 80° S extending into the Pleistocene and is
also located within one of the highest accumulation regions within interior East
Antarctica (Casey et al. 2014). This provides the unique opportunity to develop the most



highly resolved ice core record from interior East Antarctica. The South Pole is located at
an elevation of 2835 m (Casey et al. 2014) and has a mean annual temperature of -50°C
(Lazzara et al. 2012).  The high accumulation rate at South Pole (~8 cm yr$^{-1}$ snow water
equivalent, Mosley-Thompson et al. 1999; Lilien et al. 2018) relative to most of interior
East Antarctica permits glaciochemical measurements at high temporal resolution.
Occasional cyclonic events, particularly during winter months, bring seasonally variable
amounts of sea salt, dust and other trace chemicals to the South Pole (Ferris et al. 2011;
Mosley-Thompson and Thompson 1982; Parungo et al. 1981; Hogan 1997).  Due to the
favorable logistics and location at the geographic South Pole, the immediate area has
been the site of several previous ice coring campaigns (e.g. Korotkikh et al. 2014; Budner
and Cole-Dai 2003; Ferris et al. 2011; Meyerson et al. 2002; Mosley-Thompson and
Thompson 1982).  These ice cores contain records spanning the last two millennia,
providing insight into seasonal chemistry variations and background values as well as
recent snow accumulation trends.
In this paper, we focus on dating the ice itself; the dating of the gas record and the
calculation of the gas-age/ice-age difference will be the subject of a future paper.  The
procedures used to generate the data necessary for ice-core dating and the dating
techniques themselves are summarized in the remainder of the paper.
**2. Measurements and Ice core data**
***2.1 Measurements***
*2.1.1 Fieldwork and Preparation*  Drilling began at the South Pole in the 2014/2015
austral summer season at a location 2.7 km from the Amundsen-Scott station, using the
Intermediate Depth Drill designed and deployed by the U.S. Ice Drilling Program
(Johnson et al. 2014).  Drilling began at a depth of 5.10 m and reached a depth of 755 m
in January 2015. Drilling continued during the 2015/2016 season, reaching a final depth
of 1751 m.  To extend the record to the surface, a 10 m core was hand-augered near the
location of the main borehole.  Ice core sections with a diameter of 98 mm and length of
1 m were packaged and shipped to the National Science Foundation Ice Core Facility
(NSF-ICF) in Denver, Colorado.  Each meter-long section of core was weighed and
measured to calculate density and assign core depth.  The cores were cut using bandsaws
into CFA (continuous flow analysis) sticks with dimensions of 24 mm x 24 mm x 1 m
and packaged in clean room grade, ultra-low outgassing polyethylene layflat tubing
(Texas Technologies ULO) in preparation for the melter system at Dartmouth College.
An additional 13 mm x 13 mm x 1 m stick was used for water-isotope analyses at the
University of Colorado (see Jones et al., 2017 for water-isotope methods).
*2.1.2 ECM measurements* During core processing at the NSF-ICF, each core was cut and
planed horizontally to produce a smooth, flat surface (Souney et al., 2014). Electrical
conductivity measurements (ECM) were made with both direct current (DC) and
alternating current (AC). We report only AC-ECM here, as it was the primary
measurement for identifying volcanic peaks; further details are provided by Fudge et al.
(2016a). Multiple tracks were made at different horizontal positions across the core
(typically 3 tracks) and then averaged together. Measurements from each meter were


normalized by the median to preserve the volcanic signal while providing a consistent
baseline conductance to account for variations in electrode contact.
*2.1.3 Visual Measurements* Each core was examined by JF in a dark room with
illumination from below. For some cores, particularly for depths greater than ~250 m,
side-directed tray lighting using a scatter-diffuser was more effective at revealing
features. All noteworthy internal features, stratigraphy, physical properties and seasonal
indicators were documented by hand in paper log books.
Previous work at the South Pole shows that coarse-grained and/or depth-hoar
layers form annually in late summer, often capped by a bubble-free wind-crust or iced
crust up to ~1 mm thickness (Gow, 1965). We used these coarse-grained layers as the
annual "picks" (noted as late-summers). The stratigraphy in the core was generally
uniform and well-preserved, with the pattern identified by Gow (1965) continuing
downward. The depths of all noted features were recorded to the nearest millimeter. Full
details on visual layer counting are described in Fegyveresi et al. (2017).
*2.1.4. Ice Core Chemistry Analyses* Ice sticks were melted and samples collected at
Dartmouth College using a Continuous Flow Analysis – Discrete Sampling (CFA-DS)
melt system (Osterberg et al. 2006). Stick ends were decontaminated by scraping with
pre-cleaned ceramic (ZrO) knives. Cleaned sticks were then placed in pre-cleaned
holders and melted on a melt head regulated by a temperature controller in a standup
freezer. The melt head was made of 99.9995% pure chemical-vapor-deposited silicon
carbide (CVD-SIC). CVD-SIC was chosen because of its ultra-high purity, high thermal
conductivity, extreme hardness and excellent resistance to acids allowing for acid
cleaning when not in use. The melt head design includes a 16x16x3 mm high tiered and
rimmed inner section that was tapered with capillary slits to a center drain hole to
minimize the risks of contamination from outer meltwater and wicking when melting
porous firn (similar to Osterberg et al. 2006). This design provides a ≥4 mm buffer
between the exterior of each ice stick and the edge of the center tiered section. Flexible
plastic tines aligned on the four sides of the melt head keep the ice stick centered.
A peristaltic pump drew outer, contaminated meltwater away from the outer
section through four waste lines. A second peristaltic pump drew clean meltwater from
the center, tiered section of the melt head to a debubbler. The debubbler consisted of a
short section of porous expanded PTFE tubing (Zeus Aeos 0000143895) and utilized
pump pressure to force air through the tubing walls. The debubbled melt stream entered
a splitter where it was separated into three fractions: one for major ion analyses, another
for trace element analyses, and a third that passed through a particle counter and size
analyzer (Klotz Abakus), an electrical conductivity meter (Amber Science 3084), and a
flowmeter (Sensirion SLI-2000) before final collection in vials (Fig. 1). Samples were
collected in cleaned vials using Gilson FC204 fraction collectors (cleaning procedures
described in Osterberg et al. 2006). Samples were capped and kept frozen until
additional analysis.
Core depths corresponding to each sample were tracked using custom software
expanding on the concept of depth-point tracking developed by Breton et al. (2012).
Simply, software tracks each depth point in the core as it progresses through the CFA-DS
system until it reaches each collection vial. This is accomplished by using a combination



of melt rate, flow rates, and system line volumes.  Melt rates were measured with a
weighted rotary encoder tracking displacement as the ice stick melts.  Flow rates were
measured by either an electronic flow meter or by calibrating the volume per revolution
of each peristaltic pump tubing.  Fraction collector advancements were made
automatically based on melt rate, ice density (in firn), and the required sample volume
and frequency.  In addition, the software collected data from the inline particle counter
and electronic conductivity meter.  This system is capable of producing high-resolution,
ultra-clean samples and has been used successfully in previous studies (e.g. Osterberg et
al. 2017; Winski et al. 2017; Breton et al. 2012; Koffman et al. 2014).  Samples
corresponding to the top and bottom of each stick were assigned depths equal to the top
and bottom depths measured at NSF-ICF, with intervening samples scaled linearly by the
ratio of the NSF-ICF core lengths over the lengths measured by the depth encoder.  This
ensures that our data remain consistent with other SPICEcore datasets and there is no
possibility of drift due to scraping core breaks, measurement or encoder errors.
Discrete ion chemistry samples were collected every 1.1 cm on average for the
upper 800 m (Holocene) portion of the core and every 2.4 cm on average for older ice.  In
total, 112,843 samples were collected and analyzed using a Thermo Fisher Dionex ICS-
5000 capillary ion chromatograph to determine the concentrations of the following major
ions: nitrate, sulfate, chloride, sodium, potassium, magnesium and calcium.  Liquid
conductivity, particle concentration, and particle size distribution measurements were
taken continuously with an effective resolution of 3 mm.

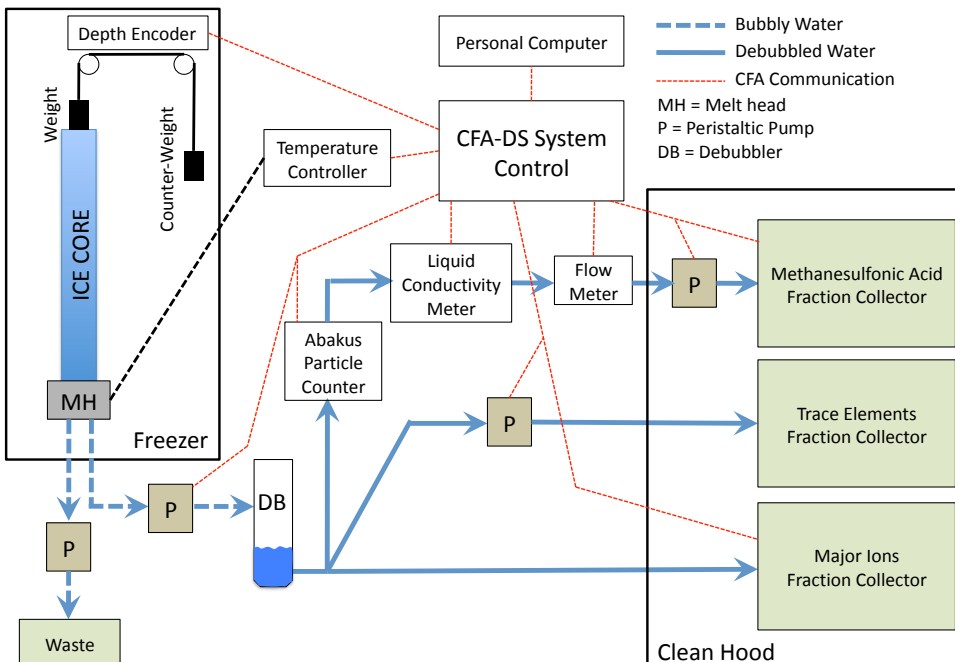

**Figure 1: A schematic representation of the Dartmouth ice core melter system.**

### 2.2 Chemistry Characteristics of SPICEcore

Previous research at the South Pole has shown that major sea salt ions ($Cl^-$, $Na^+$,
$Mg^{2+}$) have winter maxima and summer minima when compared with the position of
summer depth hoar layers (Cole-Dai and Mosley-Thompson 1999; Ferris et al. 2011).
The same conclusion was reached through comparisons with seasonal isotopic
fluctuations: sodium and magnesium peaks coincide with seasonal water-isotope minima
(Legrand and Delmas 1984; Whitlow et al. 1992). These observations are consistent with
sea salt aerosol measurements collected at the South Pole that demonstrate large sodium
influx during winter months (Bodhaine et al. 1986; Bergin et al. 1998).  The same
seasonal pattern of sea salt deposition has been observed in Holocene strata of the WAIS
Divide ice core (Sigl et al. 2016) and in other Antarctic ice cores (Kreutz et al. 1997;
Curran et al. 1998; Wagenbach et al. 1998; Udisti et al. 2012).  In the uppermost firn,
seasonal chemistry is also influenced by the operation of South Pole station and its
associated logistics (Casey et al. 2017).
In SPICEcore, sampling resolution is sufficiently high to consistently detect
annual cyclicity in glaciochemistry throughout the Holocene.  Clear annual signals are
present in several glaciochemical species to a depth of 798 m (approximately 11341 BP),
with the most prominent in sodium and magnesium (Figs. 2-3), which covary (r = 0.95; p
< 0.01) and have coherent annual maxima and minima.  Sulfate, chloride, AC-ECM,
liquid conductivity, particle count and visual stratigraphy all exhibit discernable annual
cyclicity.
The South Pole has long been recognized as a favorable location for identifying
volcanic events, reflected by previous work on South Pole paleovolcanism (Ferris et al.
2011; Delmas et al. 1992; Budner and Cole-Dai 2003; Cole-Dai et al. 2009; Baroni et al.
2008; Cole-Dai and Thompson 1999; Palais et al. 1990).  Volcanic events in SPICEcore
are evident as peaks in sulfate and ECM rising well above background values.  Within the
Holocene, the median annual sulfate maximum is 60 ppb. This background level
increases deeper in the core to values as high as 131 ppb between 18-26 ka BP, despite
the lack of annual resolution during the Pleistocene.  In contrast, sulfate concentration in
volcanic events regularly exceeds 200 ppb with occasional concentrations as high as 1000
ppb for very large signals.  For example, the pair of eruptions in 135 and 141 BP (1815
and 1809 CE), attributed to Tambora and Unknown in previous Antarctic studies
(Delmas et al. 1992; Cole-Dai et al. 2000; Sigl et al. 2013) have peak sulfate
concentrations of 518 and 281 ppb respectively, emerging well above seasonal
background values of 60 ppb.

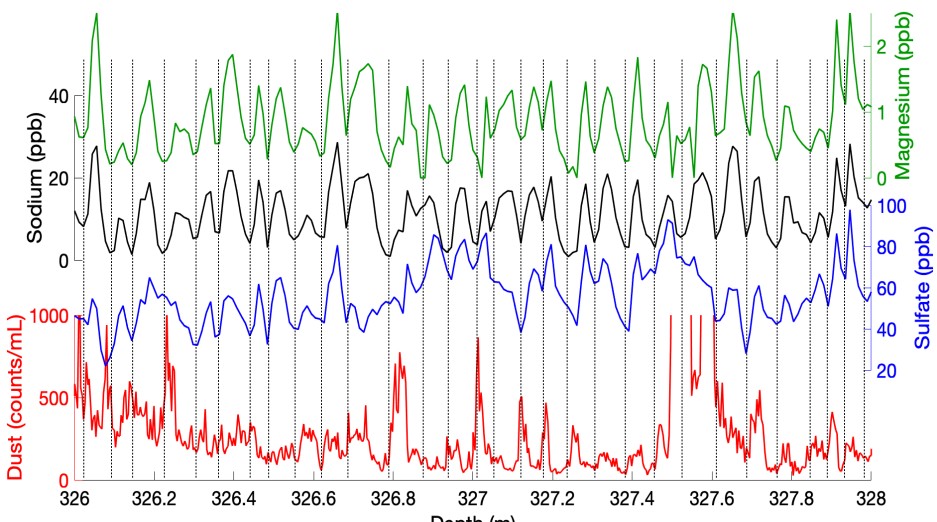

**Figure 2: Example of annual layering in a representative segment of SPICEcore. Depicted**
**are magnesium (green) and sodium (black) concentrations showing nearly identical**
**variations and clear annual cyclicity. Sulfate (blue) has consistent but less pronounced**
**layering, and dust (red; 1 micron size bin) has occasionally visible annual layering. Vertical**
**dashed lines show annual pick positions based on the data shown.**

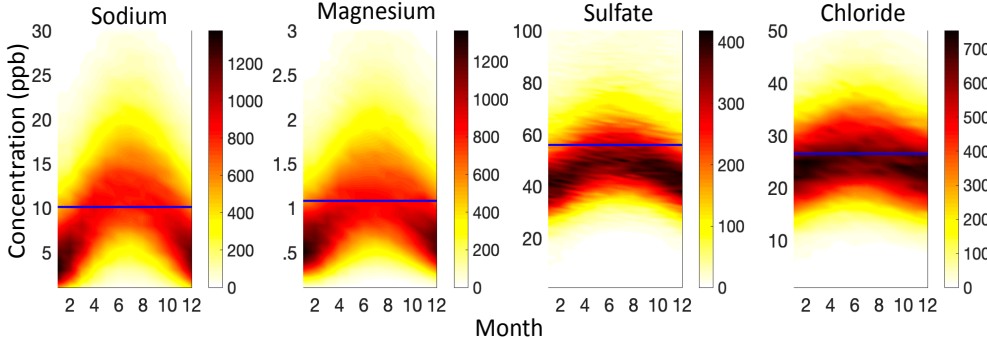

**Figure 3: Seasonal variation in magnesium, sodium, sulfate and chloride ion concentration**
**in SPICEcore from -42 to 11341 BP (11383 total years). In each panel, the horizontal axis is**
**month of the year (with 0 being Jan. 1st) from linear interpolation between mean sample**
**depth and the timescale. The vertical axis is concentration (ppb). The color scale indicates**
**the density of measurements within gridded month and concentration bins. Concentration**
**bin widths are 1 month (without claiming 1 month precision) and 1 ppb except for**
**magnesium which is 0.1 ppb. The Holocene mean concentration of each ion is shown as a**
**blue bar. Strong annual cyclicity is apparent in sodium and magnesium data. Annual**
**cyclicity is weaker in sulfate and chloride data.**



## 3. SPICEcore Dating Methods

### 3.1 Approach

The SPICEcore timescale (SP19) was developed by combining annual layer
counting with volcanic event matching between SPICEcore and the WAIS Divide
chronology. We identified 251 volcanic tie points that are clearly visible in both
SPICEcore and WAIS Divide (Sigl et al. 2016). These tie points link SP19 with the
WAIS Divide chronology, resulting in one of the most precisely dated interior East
Antarctic records. Above 798 m, ages are interpolated between volcanic tie points using
layer counts. Below 798 m, ages are interpolated between tie points by finding the
smoothest annual layer thickness profile (minimizing the second derivative) that satisfies
at least 95% of the tie points (following Fudge et al. 2014).

Although it is possible to create an independent, annually layer counted
SPICEcore timescale during the Holocene, we linked the entire SP19 chronology to the
WAIS Divide chronology for several reasons: (1) annual layers are insufficiently thick
below 798 m (approximately 11341 BP) to consistently resolve individual years,
requiring synchronization to another ice core to achieve the best possible dating accuracy.
Tying the entire SP19 chronology to the WAIS Divide core ensures consistent temporal
relationships between these two records; (2) although annual layers are remarkably well-
preserved in SPICEcore chemistry, WAIS Divide has a higher accumulation rate (Banta
et al., 2008; Fudge et al., 2016b; Koutnik et al. 2016) and stronger seasonality in
chemical constituents (Sigl et al. 2016), producing more robust annual layering (Figure
4); (3) it is expected that some years at South Pole experience very low accumulation,
resulting in a lack of an annually resolvable record during those years (Hamilton et al.
2004; Van der Veen et al. 1999; Mosley-Thompson et al. 1995, 1999); (4) an attempt to
independently date the Holocene annual layers created drift of several percent at
stratigraphic tie points. We therefore elected to anchor the SP19 timescale to WD2014,
and use the annual layer counts as a means of interpolating between WD2014 tie points
during the Holocene. The SP19 timescale spans -64 BP (2014 CE) to 54,302 +/- 519 BP,
with the annually-dated Holocene section of the core extending to 11341 BP (798 m
depth).

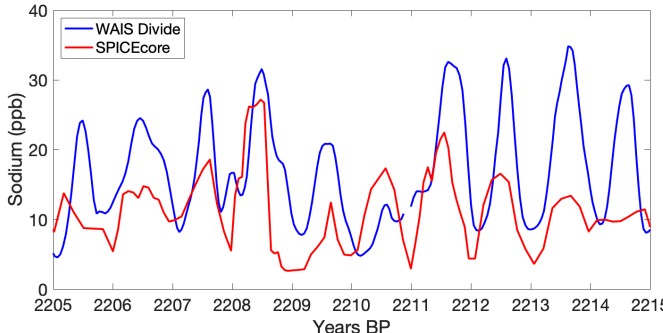

**Figure 4: Annual layering of sodium in WAIS Divide (blue) and SPICEcore (red). Annual
layers in sodium are clear in both records but are more pronounced at WAIS Divide for
most years.**





### *3.2 Procedure for identifying matching events*

The matching of volcanic events in sulfate and ECM records is commonly used to synchronize ice core timescales (e.g. Severi et al., 2007, 2012; Fujita et al., 2015), including the recent extension of the annually-resolved WAIS Divide timescale to East Antarctic cores (Buizert et al., 2018). Volcanic matching is based on the depth pattern of events more than the magnitude of the events because the magnitude in individual ice cores can vary significantly across Antarctica depending on the location of the volcano and atmospheric transport to the ice core site. The volcanic matching between SPICEcore and WAIS Divide is based primarily on the sulfate record for SPICEcore and the combined sulfur and sulfate records for WAIS Divide (Buizert et al. 2018). AC-ECM from SPICEcore and WAIS Divide was used as a secondary data set and to fill small data gaps in the sulfate record.  An example of the four data sets is shown in Figure 5. The volcanic matches were performed independently by two interpreters (TJF and DF) and then reconciled by one (TJF) with concurrence from the other (DF).  The position of each match was defined as the inception of the sulfate rise in order to most consistently reflect the timing of the volcanic event itself. Of the final 251 tie points, 229 were identified in the sulfate data by both interpreters. Of the remaining matches, 14 were made by one interpreter in the sulfate data, and at least one interpreter in the ECM data. One of the other matches was made only with ECM because of a gap in the sulfate data for SPICEcore. The last 7 matches were part of sequences not initially picked by one interpreter but deemed to be sufficiently distinct from the other events in the sequence to be included.

We note that the purpose of the volcanic matching was to develop a robust SPICEcore timescale, not to assess volcanic forcing. Thus, there are many potential volcanic matches that were not included either because they did not have the same level of certainty as the final 251 matches, or because they were in close proximity to the final matches and thus did not provide additional timescale constraints.

For the pre-Holocene section of the core, ages between the volcanic matches are interpolated by finding the smoothest annual layer thickness by minimizing the second derivative (Fudge et al., 2014). The goal of finding the smoothest annual layer thickness time series is to prevent sharp changes affecting the apparent duration of climate events on either side of a volcanic match point. The method allows the ages of the volcanic matches to vary within a threshold to produce a smoother annual layer thickness interpolation. The degree of smoothness was set such that 95% of the tie points are shifted by 1-year or less, which is a reasonable uncertainty on the precision of the volcanic matches.

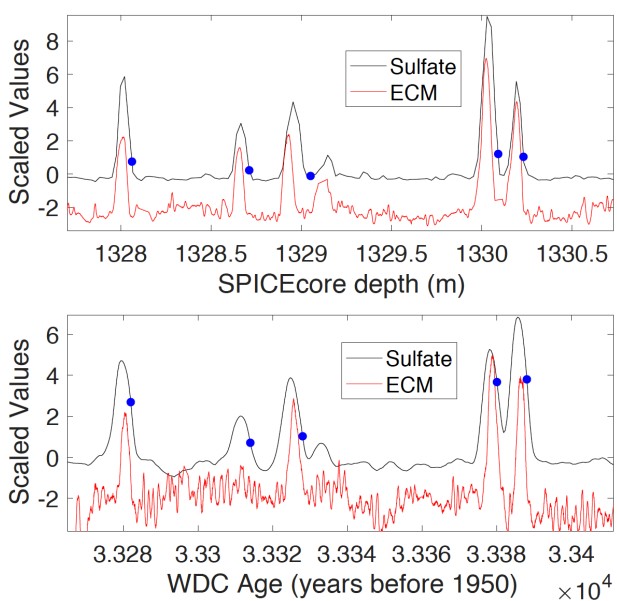

**Figure 5: An example of volcanic matching between SPICEcore (top) and WAIS Divide (bottom). Sulfate (black) and electrical conductivity (ECM; red) are shown for both ice cores. Here, five events are shown that link specific depths in SPICEcore to known ages in WAIS Divide. The position of the tie points is chosen at the beginning of the event (blue circles). The y-axis values are scaled for ease of visualization and do not indicate absolute measurement values.**

### 3.3 Annual Layer Interpretation

Annual layer counting in SPICEcore was initially done independently of the volcanic matching with WAIS Divide. To minimize and quantify timescale uncertainty, five interpreters performed the layer counting independently: DW, DF, TJF, JF, and TC. Sodium and magnesium were the primary annual indicators, but electrical conductivity, dust concentration, sulfate, chloride and liquid conductivity were also helpful in delineating individual years. To remain consistent, each interpreter agreed to place the location of Jan. 1[st] for each year at the sodium/magnesium minimum, consistent with previous interpretation of South Pole sea salt seasonality (e.g. Ferris et al. 2011; Bergin et al. 1998). Two examples of annual layering including the Jan. 1[st] positions picked by each interpreter are shown in Figure 6. Shown here are sections of high (A) and low (B) agreement among the five interpreters.

This procedure resulted in five independent timescales to a depth of 540 m, containing between 6529 and 6807 years. The details of reconciling the five independent sets of layer counts are described in the Supplemental Information. Below 540 m, only one author (DW) continued with the layer counting once the decision to use the annual layers to interpolate between volcanic events had been made. The layer counting procedure resulted in an annually resolved timescale, fully independent of any external constraints, to a depth of 798.



Above 798 meters, 86 volcanic tie points were identified, producing 85 intervals
within which a known number of years must be present.  To make the layer-counted
timescale consistent with these tie points, years were added or subtracted, as necessary,
within each interval such that the layer-counted timescale passed through each tie point
within +/- 1 year of its age, linking SPICEcore with the WAIS divide chronology.
Procedural details for adding and subtracting layers by interval are discussed in the
Supplemental Information.  In most intervals, few years needed to be added or subtracted,
with the average change in years equal to 5.6% of the interval length (Holocene intervals
ranged from 6 to 747 years).  In certain sections layer counting consistently differed from
the WAIS-tied timescale. The most notable example is from 228 to 275 m depth where
105 years (14%) needed to be added.

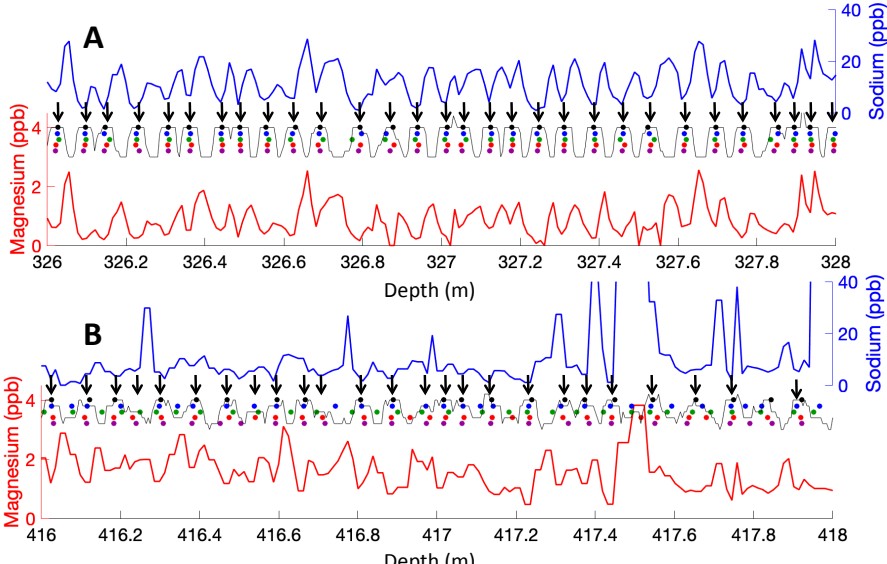

**Figure 6: Representative sections of annual layer pick positions compared with magnesium**
**(red) and sodium (blue) concentrations.  Each interpreter is represented with a different**
**color circle.  Certain sections have excellent agreement among interpreters making**
**reconciliation trivial (A), whereas other sections have poorly defined annual signals and**
**associated disagreement among interpreters (B).  The black line depicts the sum of all picks**
**within +/- 2 cm; black arrows depict the final positions of the reconciled Jan. 1st annual**
**layer picks.**

## 4. Results and Discussion

### *4.1 Characteristics of the Timescale*
The SP19 chronology extends from 2014 CE (-64 BP) at the surface to 54302 BP
at 1751 m depth.  The timescale and volcanic tie points are depicted in Figure 7 with
volcanic tie points pinning the timescale also shown.  Annual layer thicknesses near the
surface are roughly 20 cm thick (owing to the low density of firn), decreasing rapidly to
~8 cm/yr by the firn-ice transition.  The timescale is annually resolved between -64 and



11341 BP, below which resolution varies based on the distance between tie points. Using
the methods in section 3.2 (Fudge et al. 2014), we report timescale values interpolated at
10-year resolution.  The longest distance between tie points is 2476 years between 16348
and 19872 BP.

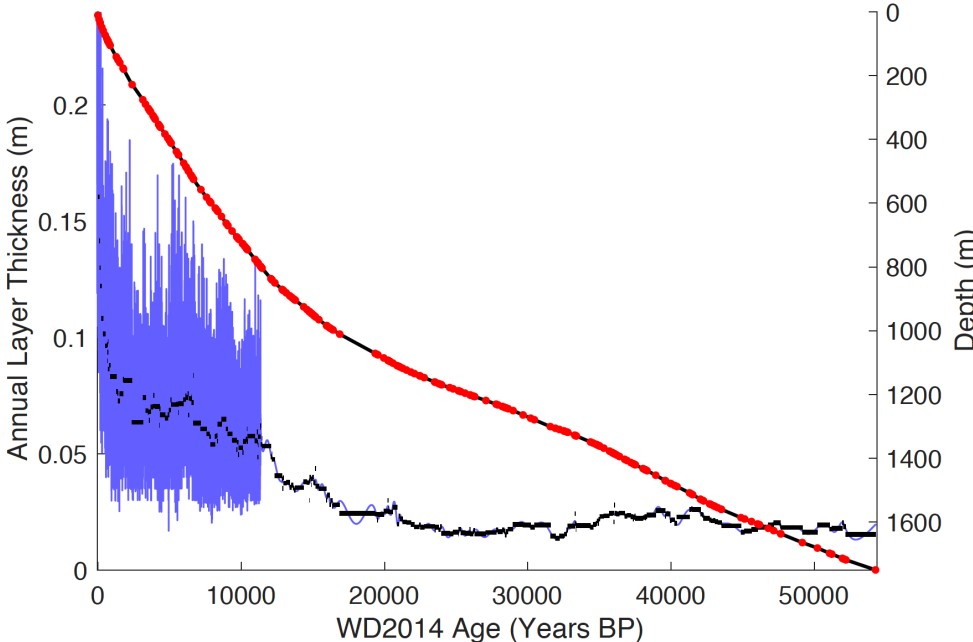

**Figure 7: The SP19 timescale, layer thickness and accumulation rate.  The SP19 depth-age**
**relationship (right y-axis, black line) is constrained by volcanic events (red dots) extending**
**to 54302 BP.  Annual layer thicknesses (left y-axis, blue) are shown at annual resolution**
**during the Holocene and as decadally-interpolated thicknesses based on the smoothest**
**annual layer thickness method (Fudge et al. 2014) during the Pleistocene.  The average**
**annual layer thickness during each volcanic interval is shown in black for comparison.**
*4.2 Uncertainties*
In discussing uncertainty values for SP19, the reported values are uncertainty
*estimates* rather than rigorously quantified 1σ or 2σ values.  There are several reasons for
this: 1) the chemicals used to count annual layers have similar cyclicity and are not
independent; 2) while each of the five interpreters counted layers independently, they
were likely employing similar strategies; 3) certain years may not be well-represented in
the data, providing insufficient information for accurate dating or quantifying
uncertainty; 4) volcanic events were identified in clusters such that each event is not
necessarily independent; 5) it is difficult to assign a numerical index of confidence to
specific volcanic tie points.  Instead, we discuss timescale uncertainties as uncertainty
estimates, which are intended to approximate 2σ uncertainties but cannot be precisely
defined as such.  This approach follows that of Sigl et al. (2016).
We assess the SP19 timescale uncertainty with respect to the previously published
WD2014 timescale (Sigl et al. 2016; Buizert et al. 2015).  The absolute age uncertainty





will always be equal to or greater than the uncertainty already associated with WD2014
(Buizert et al. 2015; Sigl et al. 2016; Fig. 8). In addition to the uncertainty in WD2014,
there is also uncertainty in our ability to interpolate between stratigraphic tie points.
During the Holocene, our layer-counting of sodium and magnesium concentration
improves the timescale accuracy between tie points. Interpolation uncertainty can be
estimated using the drift among the five different interpreters.  We calculate the number
of years picked by each interpreter in running intervals of 500 years in the final WD2014
synchronized timescale.  Under ideal conditions, each interpreter would also pick 500
years within each interval, but on average the number of years picked by interpreters
differs from the final timescale by 6.7%, usually by undercounting.  This is similar to the
metric described in section 3.3, wherein the average change in years needed to reconcile
the layer counts and volcanic tie points was 5.6% of the interval length.  Here, we report
the larger and more conservative value of 6.7%. If our layer counting skill drifts by +/-
6.7% while unconstrained by volcanic tie points, then the interpolation uncertainties
remain within +/- 18 years of WAIS Divide throughout the Holocene with the exception
of a poorly-constrained interval between approximately 1800-3100 BP.  The maximum
uncertainty within the Holocene is +/- 25 years, occurring at roughly 2750 BP, where the
nearest tie points are 373 years away at 2376 and 3123 BP.  This relationship can be
applied across the Holocene, with layers accumulating an uncertainty value equal to 6.7%
of the distance to the nearest tie point (Fig. 8; blue).
Below the Holocene (798 m depth), there were no annual layers to aid in our
interpolation of the timescale, leading to larger uncertainties.  Our assumption of the
smoothest annual layer thickness (Fudge et al. 2014) satisfying tie points is the most
accurate interpolation method in the absence of additional information, at least in
Antarctic ice (Fudge et al. 2014).  Using the WAIS Divide ice core as a test case, Fudge
et al. (2014) estimated that the interpolation method accumulates uncertainties at a rate of
10% of the distance to the nearest tie-point, roughly 50% faster than the uncertainty of
periods with identifiable annual layers.  The longest interval with no volcanic constraints
is between 16348 and 19872 BP.  At 18110 BP, the center of the interval, the
interpolation uncertainty reaches a maximum of 124 years, although uncertainties are
proportionally lower in other intervals with closer volcanic tie points.
Figure 8 shows the total uncertainty estimates associated with the SP19
chronology, with interpolation uncertainties added to the published WAIS Divide
uncertainties.  The WD2014 and interpolation uncertainties are added in quadrature since
the two sources of uncertainty are independent.  The maximum estimated uncertainty in
SP19 is 533 years at 34050 BP, the majority of which is attributed to uncertainties in
WD2014. While it is not possible to rigorously quantify uncertainties throughout SP19,
we believe these estimates provide reasonable and conservative values suitable for most
paleoclimate applications.  We acknowledge there is additional uncertainty related to the
accuracy of our assigned stratigraphic tie points.  Because of the conservative procedures
discussed in section 3.1 wherein only unambiguous matches were used in linking the
WAIS Divide and SPICEcore timescales, it is unlikely that any of these matches are in
error.  In previous work (Ruth et al. 2007), potential errors associated with tie points have
been estimated by removing each tie point one at a time, and interpolating between the
new series of tie points (with one point missing). If this procedure is repeated for each tie
point and for each depth, the maximum error in age resulting from the erroneous



inclusion of a tie point is approximately 83 years. However, because clusters of volcanic
events were used to match the WAIS Divide and SPICEcore records, each tie point is not
necessarily independent. Therefore, this method is more useful at sections of widely
spaced tie points with greater potential uncertainties, but underestimates the uncertainties
surrounding closely spaced events in SPICEcore and WAIS Divide.

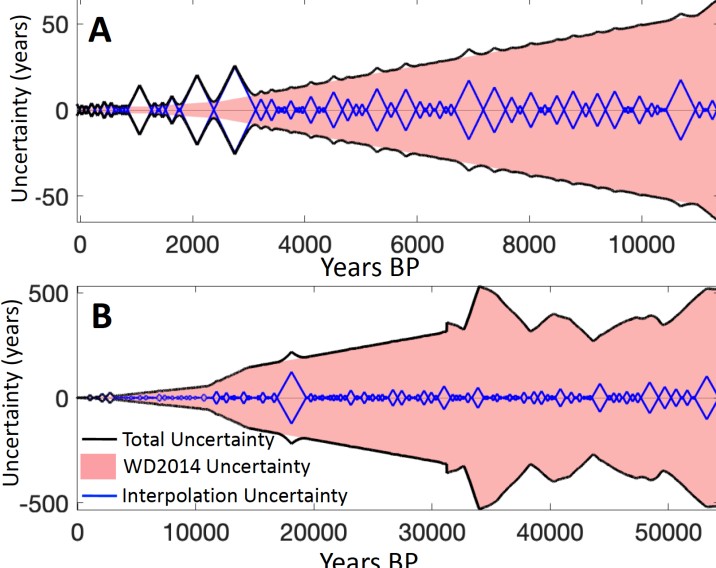

**Figure 8: Uncertainty estimates in the SP19 timescale. The pink shading indicates the**
**published uncertainty associated with the WAIS Divide timescale. The blue lines indicate**
**the estimated uncertainty due to interpolation by layer counting (Holocene) and by finding**
**the smoothest annual layer thickness history (Fudge et al. 2014; Pleistocene). Total**
**uncertainty (black) is defined here as the root sum of the squares of the interpolation and**
**WD2014 uncertainties. Total uncertainty estimates remain within +/- 50 years for most of**
**the Holocene (A), but are as high as 533 years in the Pleistocene (B).**
*4.3 Comparison with Visual Stratigraphy*
Visual stratigraphy in SPICEcore provides an independent check on the
glaciochemical layer counting we used to interpolate the Holocene depth-age scale
between tie points. Visual layer counting was conducted to a depth of 735 m (~10,250
years BP; Fegyveresi et al. 2017). We calculate the offset between the visual stratigraphic
timescale and a linear interpolation between tie points and do the same for the chemistry
layer counts (Fig. 9). If both the chemical and visual layer counting methods are
capturing the true variability in layer thickness within intervals, then both would show the
same structure within each interval.
There is broad correspondence between visual and chemical stratigraphy at all
depths, which, with their almost completely independent origin and measurements
techniques, is highly reassuring. In detail, though, there is little high-frequency
correspondence between visual and chemical layer counts below 1400 BP (150 m depth),
although a direct comparison is not possible since visible layer counts were not linked to
stratigraphic tie points between 1400-2400 BP and 8400-9500 BP.  Furthermore, visible
layer counts were matched to the tie points within error of the WAIS Divide timescale,
whereas the chemistry layer counts were forced to match within +/- 1 year of each tie
point.  In counting visible layers, occasional under- and overcounting of depth hoar layers
within annual strata is likely, especially in deeper ice where thinning will make adjacent
layers appear even closer.  There were some intervals (e.g. 2000 – 2500 BP) in the core
that appeared more homogeneous during viewing, and therefore annual layer choices
have a higher level of uncertainty.   Because of the differences between methodologies in
matching to tie points and because of the uncertainties in visual counting below 2000 BP
(200 m), we did not attempt to reconcile the visible and chemical layer counts, but
instead rely only on the annual layers in the chemistry data.
Between 100 and 1400 BP, both visible and glaciochemical timescales remain
remarkably coherent and do not indicate drift of more than +/- 2 years.  Over this interval,
the correlation between the visible and chemical layer offsets from constant annual layer
thickness (red and blue curves in Figure 9) is 0.74. The correlation between the two layer
counting methods is as high as r = 0.85 between the tie points at 841 and 1268 BP. The
discrepancy within the top 100 years is due to the tie point at 10.58 m, which was not
included at the time of visible layer counting, as well as low layer chemical counting
confidence within the firn column. There is no obvious relation between the
accumulation rate and statistical agreement among methods.

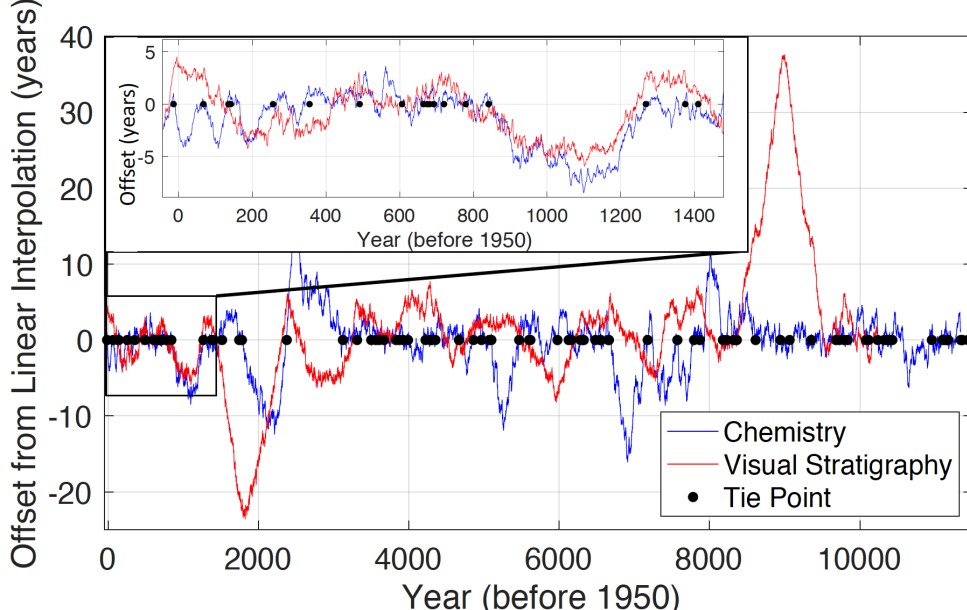

**Figure 9: Comparison between visible layer (red) and chemistry-based (blue) Holocene**
**annual timescales.  Both curves are shown as residual values with respect to a linear**
**interpolation between tie points (black circles).  When the shape of the red and blue curves**
**is similar between tie points, we infer relatively high accuracy in both methods.  The region**
**showing the closest agreement between methods is shown in the inset with both curves**
**remaining within 2 years of each other despite a long section with no tie points (841 to 1286**
**BP).**





### *4.4 Accumulation Rate History*

The SP19 timescale allows us to produce annually-resolved estimates of past snow accumulation to 11341 BP (Fig. 10). We apply a Dansgaard-Johnsen model (Dansgaard et al. 1969) to estimate the amount of thinning undergone by each layer of ice. Since the entirety of the Holocene in SPICEcore is located within the top third of the core (over 1900 m above the bed), the challenges associated with reconstructing surface accumulation are smaller than at sites with records closer to the bed (e.g. Kaspari et al. 2008, Thompson et al. 1998, Winski et al. 2017). Radar measurements indicate a bed depth at the South Pole of 2812 m, giving an ice-equivalent thickness of 2774 m, using the South Pole density function developed by Kuivinen et al. (1982). We used a kink height of 20% of the ice thickness and an input surface accumulation rate of 8 cm/yr, consistent with the parameters used by Lilien et al. (2018). The average Holocene accumulation rate is 7.4 cm/yr, in excellent agreement with results of previous studies (Hogan and Gow 1997; 7.5 cm/yr to 2000 BP; Mosley-Thompson et al. 1999 – 6.5-8.5 cm/yr for late 20[th] century). The upstream flow dynamics are too complicated for a static 1-D model to accurately determine the thinning function before the Holocene.

As discussed in Lilien et al. (2018), Koutnik et al. (2016), and Waddington et al. (2007), South Pole layer thicknesses are affected by 1) spatial variability in surface accumulation being advected to South Pole; 2) past climate-related changes in snow accumulation; and 3) post-depositional thinning due to ice flow. Thinning models can account for only the third factor. Understanding of Holocene climate history as recorded at other sites and in other indicators in SPICEcore, combined with knowledge of the modern upglacier variation in accumulation (Lilien et al., 2018), make it clear that the Holocene SPICEcore time-variations in accumulation are primarily from advection of spatial variations. Figure 10 shows Holocene accumulation rate in SPICEcore (black) compared with geophysically derived accumulation estimates over space using ice-penetrating radar (blue, details in Lilien et al. 2018). Using the present-day surface velocity field and the inferred 15% increase in flow rate, present day upstream surface accumulation rates were matched with corresponding ages at the SPICEcore borehole (Lilien et al. 2018). The close match between present-day near-surface accumulation rates upstream and the annual accumulation rate in SPICEcore shows that the millennial-scale signal of accumulation rate in SPICEcore is related to spatial patterns of snow accumulation upstream of South Pole.

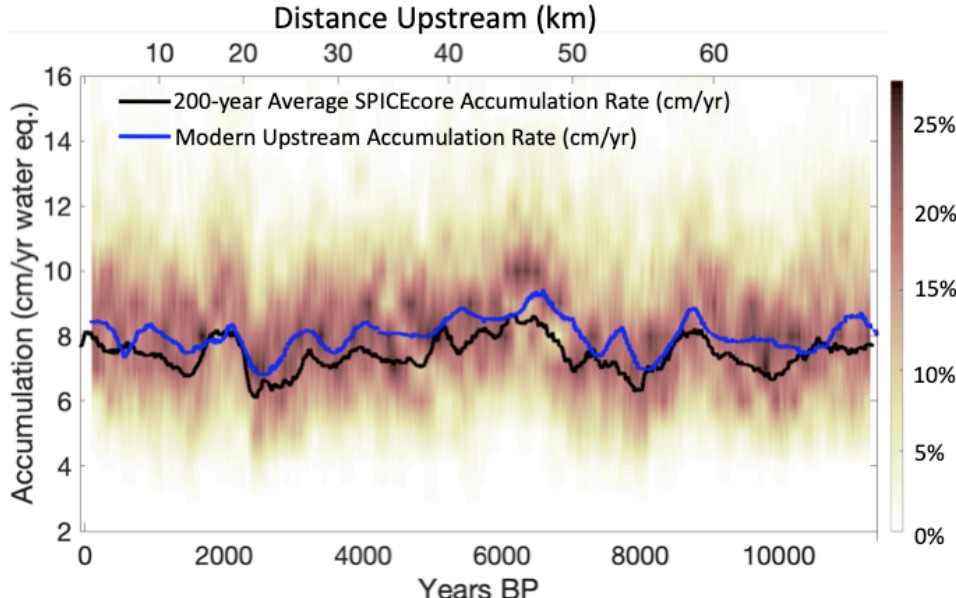

**Figure 10: The Holocene accumulation rate history in SPICEcore. Shading indicates a running histogram of accumulation rate with darker colors indicative of more years at a given accumulation rate. The color axis (left) indicates percentage of years with a given accumulation rate within 1 cm accumulation bins across 200-year sliding intervals. The solid black line is the 200-year running mean of accumulation rate. These data are compared with modern spatial accumulation rates upstream of SPICEcore (blue; upper x-axis; Lilien et al. 2018).**

A striking feature in the Holocene accumulation record in SPICEcore is the sharp dip centered on 2400 BP. Annual layers were notably less clear in that portion of SPICEcore because low accumulation rates led to low sampling resolution (5-6 samples/year). For instance, in the interval between 228-275 m, the interpreters picked between 511 and 670 years, when 747 years are present based on the volcanic tie points. The cause of the sharp drop in accumulation is not clear. Modern accumulation rates upstream of SPICEcore were measured using a 20 m-deep isochron imaged with ice penetrating radar (Lilien et al. 2018). These results show lower accumulation in the location where the 2400 BP ice originated (Fig. 10). However, the modern upstream spatial pattern of accumulation shows a decline that is both more gradual and less than half the magnitude of the 2400 BP change in SPICEcore. It is possible that this represents a climatic signal, but we note sharp accumulation variations at this time that are not observed in the WAIS Divide core (Fudge et al. 2016b; Koutnik et al. 2016). Instead, we hypothesize that this event was most likely a transient local accumulation anomaly. Farther upstream at ~75km from South Pole, there is an accumulation low where the rate of change is approximately 3 cm/yr in 2 km. With the current South Pole ice flow velocity of 10 m/yr, this could explain a 3 cm/yr decrease in 200 years, similar to what is observed at 2400 BP. If a climate-driven accumulation anomaly did contribute to this sharp change, these anomalies do not appear to be common, as we see no other large and sustained change in the annual timescale.



On sub-centennial timescales, the effects of upstream advection of spatial
accumulation patterns are likely smaller, such that annual-to-decadal patterns in snow
accumulation in SPICEcore may be indicative of climate conditions.  Previous studies
have used a snow stake field 400 m to the east (upwind) of South Pole station to assess
recent trends in accumulation rate with differing results.  Mosley-Thompson et al. (1995,
1999) found a trend of increasing snow accumulation during the late 20[th] century, while
Monaghan et al. (2006) and Lazzara et al. (2012) found decreasing snow accumulation
trends between 1985-2005 and 1983-2010, respectively.  No significant trends exist in the
SPICEcore accumulation record within the last 50 years, although there is a significant (p
= 0.046) increasing trend in snow accumulation in SPICEcore since 1900.  Note that
errors in measured firn density would influence this accumulation trend.
*4.5 Nitrate Variability, $\delta^{15}N$ of $N_2$, and Accumulation*
SPICEcore nitrate concentrations provide independent support for the Holocene
accumulation rate history implied by the SP19 timescale.  Previous studies have
recognized an association between accumulation rate and nitrate concentration among ice
core sites (Rothlisberger et al. 2002).  Nitrate in surface snow, exposed to sunlight, results
in photolytic reactions that volatilize nitrate and release it to the atmosphere (Erbland et
al. 2013, Grannas et al. 2007; Rothlisberger et al. 2000). Evaporation of $HNO_3$ may also
significantly contribute to nitrate loss in the surface snow (Munger et al. 1999; Grannas et
al. 2007).  Under low-accumulation conditions such as in East Antarctica, the amount of
time snow is exposed at the surface is the dominant control on nitrate concentration, such
that with more accumulation, snow is more rapidly buried and retains higher nitrate
concentrations (Rothlisberger et al. 2000).
There is close correspondence between accumulation rate and nitrate
concentration in SPICEcore (Fig. 11A).  This association is strongest on multidecadal to
multicentennial timescales with correlation coefficients between accumulation rate and
nitrate reaching peak values after 512-year smoothing (r = 0.60; Fig. 11 inset).  Although
the smoothing makes standard metrics statistical significance inapplicable, the similarity
between time series is expected given the previous work described above.  Among sites,
an inverse relationship exists between seasonal amplitude of nitrate concentration and
accumulation rate.  High-accumulation sites such as Summit, Greenland exhibit strong
annual nitrate layering, whereas low-accumulation sites such as Vostok (~2 cm w.e./yr;
Ekaykin et al. 2004) and Dome C (~3.6 cm w.e./yr; Petit et al. 1982) do not show annual
nitrate layers at all (Rothlisberger et al. 2000).  SPICEcore has much higher accumulation
rates than Vostok or Dome C, and retains weak seasonality in nitrate wherein nitrate
often peaks in the summer months, the mechanisms for which are complex (Grannas et
al. 2007; Davis et al. 2004).  As expected, the seasonal amplitude of nitrate over the
Holocene closely follows nitrate concentration and accumulation rate (Figure 11B) and is
even more highly correlated with accumulation than nitrate concentration itself,
especially on multicentennial to millennial timescales (r = 0.80 at 512-year smoothing).
Nitrate and accumulation rate are entirely independent variables in terms of their
measurement, adding confidence to the annual layer counting and tie points underlying
the SP19 chronology.

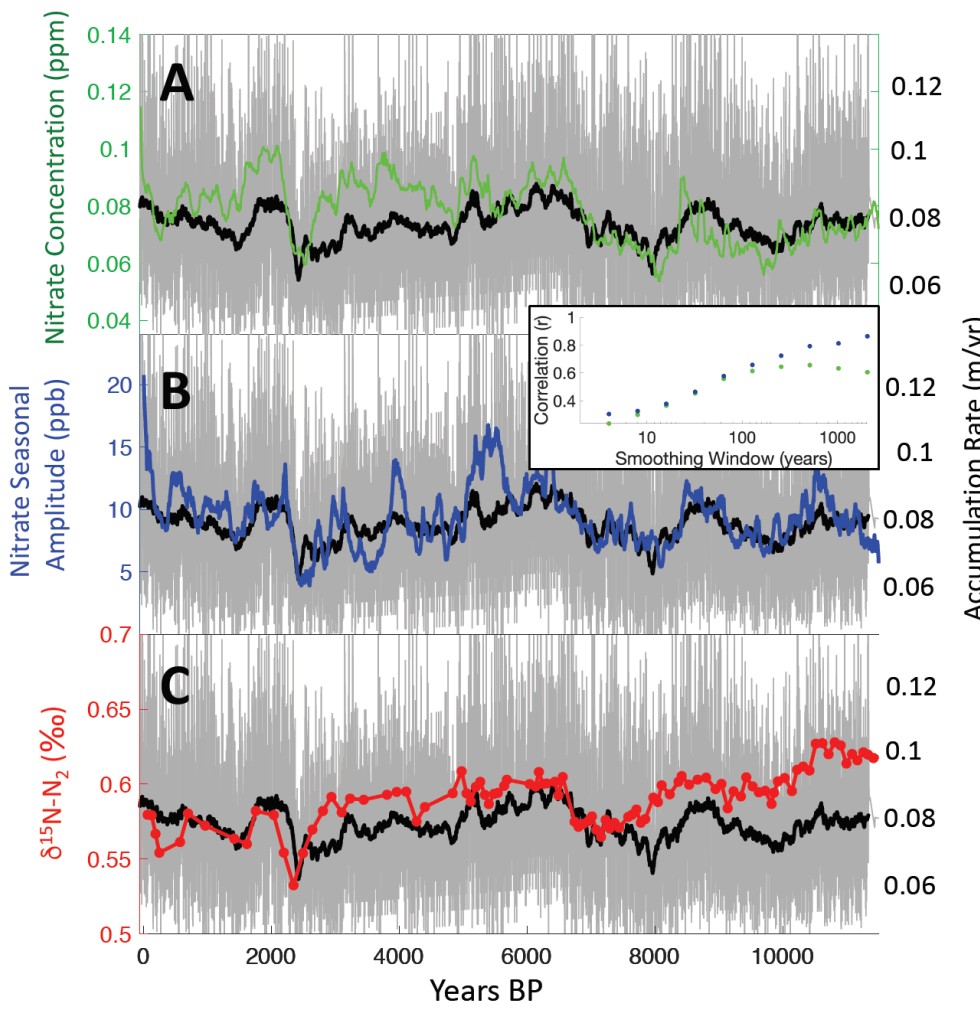

**Figure 11: The Holocene accumulation rate at the South Pole compared with nitrate and δ¹⁵N.** In each panel, annual accumulation rates are depicted in gray, with the running 100-year mean shown in black. These results are compared with 100-year median annual values of nitrate concentration (A) and seasonal amplitude in nitrate concentration (B) as well as δ¹⁵N values (C). All three metrics exhibit shared variability on multicentennial to millennial timescales. The inset shows the correlation between accumulation rate and nitrate concentration (green) from panel A, and between accumulation rate and nitrate seasonal amplitude (blue) from panel B, against length of the smoothing window, with both exhibiting high correlations, especially at lower frequencies.

The relationship between inferred variations in accumulation rate and nitrate concentration breaks down prior to the Holocene, but a relationship between nitrate and calcium concentrations emerges. During the Pleistocene, the correlation between centennial median of calcium and nitrate is $r = 0.80$ ($p < 0.01$; Figure 12), compared with $r = 0.26$ ($p < 0.01$) during the Holocene. Rothlisberger et al. (2000, 2002) observed the





same pattern at Dome C, and attributed it to the stabilization of nitrate through interaction
with calcium and dust. They proposed that $CaCO_3$ and $HNO_3$ react to form $Ca(NO_3)_2$,
which is more resistant to photolysis and consequently leads to higher concentrations of
nitrate in the glacial age snowpack despite lower accumulation rates.  The stabilization
effect of calcium apparently overtakes photolysis and evaporation of nitrate in terms of
importance only at the very high calcium concentrations as seen in the pre-Holocene ice.

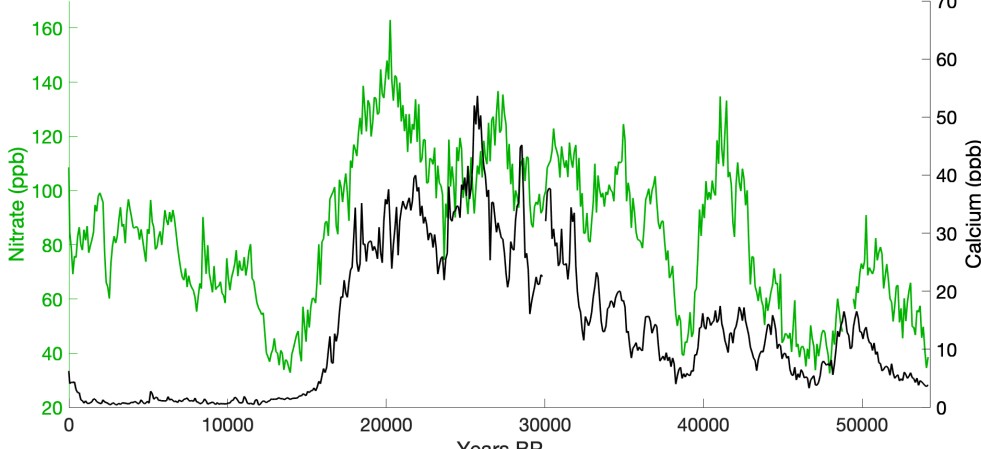

**Figure 12: Nitrate and calcium concentrations in SPICEcore.  There is low centennial-scale**
**correlation (r = 0.26; p < 0.01) between calcium and nitrate ions during the Holocene, when**
**accumulation is the dominant control on nitrate concentration (Fig. 11).  During the**
**Pleistocene, centennial median nitrate and calcium are positively correlated (r = 0.80; p <**
**0.01).**
Stable isotope ratios of atmospheric diatomic nitrogen ($\delta^{15}$N-$N_2$) in trapped air in
SPICEcore show a pattern similar to accumulation rate within the Holocene (Fig. 11C).
$\delta^{15}$N-$N_2$ values were measured using the procedures described by Petrenko et al. (2006).
The $\delta^{15}$N-$N_2$ in ice cores is driven by gravitational enrichment and is a proxy for past
thickness of the firn column (Sowers et al 1992). Firn densification rates depend
primarily on temperature and overburden pressure, with the second parameter closely
linked to the accumulation rate at the site. Low temperatures and high accumulation rates
both act to thicken the firn, thereby increasing $\delta^{15}$N-$N_2$ (Herron and Langway 1980,
Goujon 2003).
We perform a simple attribution study to see whether $\delta^{15}$N-$N_2$ variations can be
explained by reconstructed accumulation history or variable temperature. We compare
three climatic scenarios in a dynamical version of the Herron-Langway densification
model (Buizert et al. 2014). The first uses variable temperature (from $\delta^{18}$O using a
scaling ratio of 0.8‰/°C) and variable accumulation (from annual layer thickness)
forcing; a second uses constant temperature (-51.5 °C) and the variable accumulation
forcing; a third uses variable temperature and constant accumulation (7.8 cm/yr) forcing.
The correlations between the $\delta^{15}$N-$N_2$ data and each model run are displayed in Fig. 13
for both raw and detrended time series.  The model scenario forced by both temperature
and accumulation has the best correspondence with the $\delta^{15}$N-$N_2$ data (r = 0.65; p < 0.01).
While secular changes in temperature appear to be driving the decreasing trend in $\delta^{15}$N-





$N_2$, millennial-scale fluctuations in $\delta^{15}N$-$N_2$ appear to be driven by accumulation,
supported by the high correlation (r = 0.64; p < 0.01) with the accumulation-only model
run using detrended time series. In particular, a sharp drop in $\delta^{15}N$-$N_2$ is present at
approximately 2400 BP, coincident with (and driven by) the local minimum in
accumulation. These experiments provide additional confidence in the reconstructed
accumulation history. To our knowledge, these data represent the best observation of
accumulation-driven $\delta^{15}N$-$N_2$ variation, making it a valuable target for benchmarking firn
densification model performance (Lundin et al. 2017).

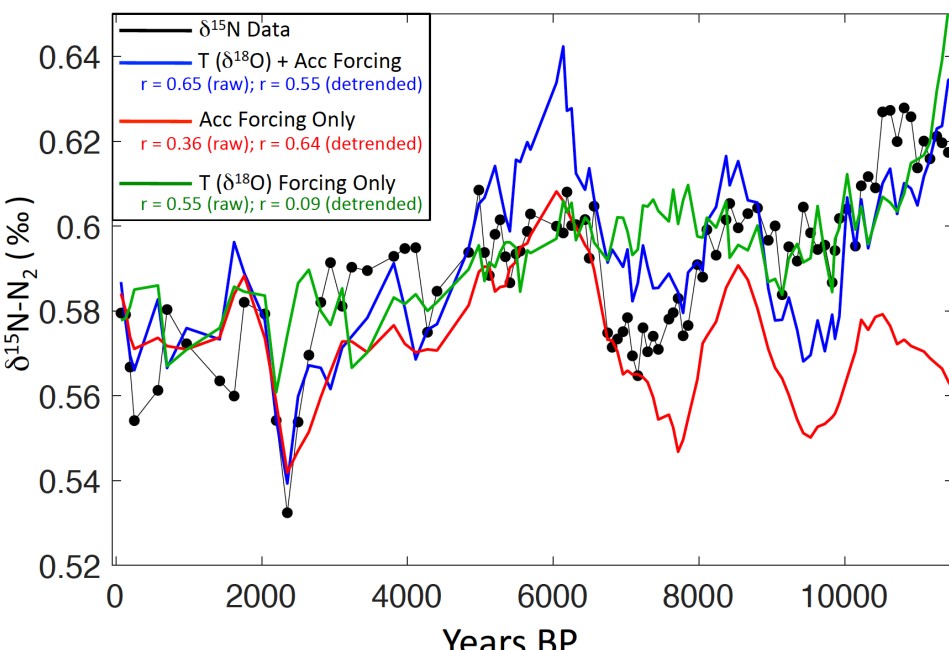

**Figure 13: Results from three firn models compared with $\delta^{15}N$ variations in SPICEcore**
**(black). The model run incorporating only $\delta^{18}O$-based temperature (green) does not**
**capture the millennial-scale variations in $\delta^{15}N$, whereas the models using only accumulation**
**(red) and both accumulation and $\delta^{18}O$-based temperature (blue) are able to reproduce the**
**observed millennial-scale $\delta^{15}N$ changes. Correlations between the $\delta^{15}N$ data and the three**
**model runs are reported in the legend with correlation coefficients calculated for both raw**
**and linearly detrended time series.**

**5. Summary**

The SP19 includes the last 54,366 (-64 to 54,302 BP) years, and is the oldest and
most well-constrained ice core timescale from the South Pole. SP19 was developed using
251 volcanic events that link the SPICEcore timescale with the WAIS Divide chronology
WD2014 (Sigl et al. 2016; Buizert et al. 2015). High-resolution chemical records in
SPICEcore during the Holocene provide the only annually resolved full-Holocene
paleoclimate record in interior East Antarctica. Within the Holocene, SP19 uncertainties

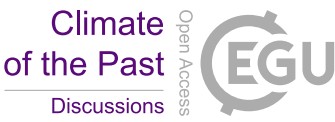

are in the range of +/- 18 years with respect to WAIS Divide, with the exception of the
interval between 1800-3100 BP when low accumulation and sparse volcanic controls lead
to uncertainties as high as +/- 25 years.  During the Pleistocene, SP19 uncertainties are
inversely related to the density of tie points, with maximum uncertainties reaching +/-
124 years relative to WD2014.  Results show an average Holocene accumulation rate of
7.4 cm/yr with millennial-scale variations that are closely linked with advection of spatial
surface-accumulation patterns upstream of the drill site.  Nitrate concentrations, nitrate
seasonal amplitude, and $\delta^{15}$N-N$_2$ variability are positively correlated with accumulation
rate during the Holocene, providing independent confirmation of the SP19 chronology.

## Competing Interests

The authors declare that they have no conflict of interest.

## Data Availability

The SP19 chronology, associated tie points, uncertainty estimates and supporting data
sets will be archived at the National Climate Data Center (www.ncdc.noaa.gov) and the
U.S. Antarctic Program Data Center (http://www.usap-dc.org) with the publication of this
paper.

## Author Roles

All authors contributed data to this study.  DW, DF, EO, JCD, ZT, KK, and NO
measured the ice core chemistry.  TJF and EDW collected the ECM data.  JF and RA
performed the visual analysis.  CB, JE, EB, RB, JF and TS made the gas measurements.
ES, EK, TJ, and VM made the isotope measurements. DW, TJF, DF, JF and TC
performed the annual layer counting.  TJF and DF performed the volcanic matching.
DW, TJF, DF, EO, JF and CB wrote the paper with contributions from all authors.

## Acknowledgements

This work was funded through U.S. National Science Foundation grants 1443105,
1141839 (Steig), 1443336 (Osterberg), 1443397 (Kreutz), 1443663 (Cole-Dai), 1443232
(Waddington, Fudge), 1142517, 1443470 (Aydin), 1443464 (Sowers), 1443710
(Severinghaus), 1542778 (Alley, Fegyveresi), 1443472, 1643722 (Brook).  We thank
Mark Twickler and the SPICEcore Science Coordination Office for administering the
project; the U.S. Ice Drill Program Office for recovering the ice core; the 109th New
York Air National Guard for airlift in Antarctica; the field team who helped collect the
core; the members of South Pole station who facilitated the field operations; National
Science Foundation Ice Core Facility for ice core processing; and the many student
researchers who produced the data underlying the SP19 timescale.



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
