# Peer review of "The SP19 Chronology for the South Pole Ice Core - Part 1: Volcanic matching and annual-layer counting"

_Climate of the Past, 2019_

## Referee Comment (RC1) · Frédéric Parrenin (Referee) · 2 Jul 2019

Winski et al. present a first chronology, named SP19, for the South Pole Ice Core (SPICECore), back to 54,302 yr BP. This chronology is based on a combination of 251 volcanic matching to the WAIS Divide ice core and annual layer counting (back to 11,341 yr BP for the latter). More precisely, the SP19 chronology is strictly tied to the WD2014 chronology at the depth of the volcanic matches, and then layer counting is used to interpolate in-between. Before 11,341 yr BP, a spline interpolation method is used instead of annual layer counting. The layer counting is based on CFA measurements of magnesium, sodium, sulfate, chloride and dust. It has been performed by

4 different operators and reconciliation is found a posteriori when there is a discrepancy. A comparison is also made with visual stratigraphy but chemical stratigraphy is preferred because it is found to be more accurate.

It is argued that the WD2014 is used because it is more precise (annual layer thickness is larger at WD) and to have a WD2014 compatible time scale. The relative uncertainty to WD2014 is small during the Holocene, generally less than 18 yr and always less than 25 yr. For older parts, the relative uncertainty to WD2014 is less than 124 yr.

The accumulation rate which is found in the SPICEcore is found to be mainly due to the upstream spatial pattern of accumulation along the flow line. The accumulation reconstruction is also compatible with nitrate concentration, nitrate amplitude of seasonal variations and N-15 of N2 in air bubbles (through a dynamical Herron-Langway firn model for the latter). It is therefore argued that the SPICEcore is a good candidate to test the influence of surface accumulation rate on the Lock-In Depth.

This article is very well written and its content is consistent. I therefore recommend to accept it.

I only have a few technical corrections.

- l . 385: suppress "accumulation rate" since it is actually not plotted on this figure.

- l. 489, l. 531 and l. 535: double space after dot.

---

## Referee Comment (RC2) · Anonymous Referee #2 · 3 Jul 2019

In the paper "The SP19 Chronology for the South Pole Ice Core - Part 1: Volcanic matching and annual-layer counting" by D. A. Winsky and co-authors a new timescale (called SP19) for the SPICEcore is presented. This new time scale was partly achieved by annual layer counting but the main guide to build the age scale is a robust volcanic match coming from a comparison with WAIS Divide ice core chronology. Given the best quality of annual layering in WAIS Divide (as shown in figure 4) I think that the authors choose the best methodological approach to build this time scale, considering WAIS as the most reliable annually counted scale. The discussion about the uncertainty of the age scale is really well done and confirms the goodness of this first SP19 age scale both for the Holocene period (with a maximum uncertainty of about 25 years) and for

older ice (maximum 124 years in the longest time window without tie points). The paper is clear and well written and I recommend its publication after considering the following minor points.

Minor comments:

Figure 5: I would recommend to change the x-axis in kyr BP instead of using x10ˆ4. In my opinion it would be much readable.

A list (table) of the volcanic horizons used to match WDC and SPICEcore would be valuable if inserted in the text or in supplementary material (even if archived at the NCDC or other repository).

Line 427: I would change this sentence to "Below 798 m depth (start of the Holocene)...."

---

## Referee Comment (RC3) · Anders Svensson (Referee) · 3 Jul 2019

Review by Anders Svensson of manuscript entitled 'The SP19 Chronology for the South Pole Ice Core – Part 1: Volcanic matching and annual-layer counting' submitted to Climate of the Past by D. Winsky et al.

The manuscript (MS) introduces a stratigraphic chronology SP19 of the South Pole SPICE ice core based on 1) Holocene layer counting in high-resolution discrete chemistry samples and continuous records, and 2) a transfer of the WD2014 chronology based on identification of 251 common volcanic match points distributed over the last 54 ka. The layer counting is compared to a previously obtained independent layer

counting from the same core based on visual stratigraphy alone. Furthermore, the authors are introducing an accumulation rate profile for the SPICE core based on a kink model. The model is compared to upstream accumulation patterns, to nitrate concentration profiles and to a d15N of N2 profile that all seem to support the obtained accumulation profile.

Overall, the MS is well written, well referenced and the figures are clear and illustrative of the study. The MS is well structured, the language is clear and the conclusions are well argued for. I only have a few comments below for the authors to consider.

The authors perform multiple careful counting of annual layers of the Holocene using chemical parameters, continuous dust and conductivity. They then compare their resulting layer counting to an independent layer counting based on visual stratigraphy alone, and find an overall good agreement between the two approaches (Figure 9). Whereas this is a good test to see how well the two independent approaches are, it would probably have resulted in a better overall time scale, if all of the available high-resolution records (chemistry + visual stratigraphy) had been combined in a common dating exercise from the beginning?

Whereas I agree to the approach of transferring the WD2014 chronology to SPICE core rather than publishing a new independent time scale for SPICE core, it still seems like quite a large effort to do 4x independent layer counting of SPICE just to end up doing a transfer of time scale? Probably most of that time scale transfer could have been done based on a depth-depth matching alone (WDC – SPICE) similar to the approach taken in Figure 5?

There are certain depth intervals (228-275m and 626-687m), where all of the independent layer counting plus the automated Straticounter dating approach consistently count significantly fewer annual layers than suggested by the transfer of the WD2014 time scale. I understand that this consistent undercounting is associated with periods of exceptionally low accumulation (upstream) at SPICE. Are the authors convinced,

however, that their layer counting is wrong and that there is not a problem with the WD2014 counting in one or both of those periods? In other words, can the authors account for all of the 'missing' layers when they go back and recount the critical sections in SPICE core? I'm not suggesting that the authors should revise the WD2014 time scale, but independent checks are always useful, and considering the effort put into precise dating of SPICE, an outcome may be suggestions for future revisions of the WDC chronology?

112,843 samples were collected and analyzed individually for this project! Did the authors consider doing fewer discrete and more continuous sample analysis? A CFA system optimized for depth/time resolution should be able to resolve the annual layering throughout the Holocene period. Of course, I'm not suggesting to do that now, but for future projects it may be an alternative?

A depth-difference relation figure between the two synchronized ice cores (SPICE, WDC) is very good for evaluating the synchronization and/or to identify regions where accumulation/thinning of the two records deviate. See Figure 2 of Seierstad et al., QSR, 2014: 'Consistently dated records from the Greenland GRIP, GISP2 and NGRIP ice cores for the past 104 ka reveal regional millennial-scale d18O gradients with possible Heinrich event imprint'.

After having nicely synchronized the SPICE and WDC records, it would be nice to see the two climate profiles (water isotopes) in the same figure on their common time scale. If for whatever reason that is not possible, maybe the Calcium profiles of the two ice cores can be shown together? It is difficult for the reader to evaluate the quality of the volcanic matching without seeing a comparison of some parameter of the two ice core records.

Minor comments:

Accumulation mistakenly used in figure 7 caption, already mentioned by Frederic. In Figure 3 is shown the seasonal variability of four impurities but not including nitrate.

In section 4.5 and in Figure 11B the nitrate seasonality is discussed. Maybe it makes sense to include the seasonal variability of nitrate in Figure 3?

In Figure 7, the annual layer thickness appears to stay constant or even increase throughout the glacial part of the record. Wouldn't one normally expect a thinning of annual layering with depth?

---

## Author Comment (AC1) · 13 Aug 2019

Dear Frédéric Parrenin,

Thank you very much for the very useful and positive comments. We greatly appreciate your input. Below are our line-by-line responses to your review.

Winski et al. present a first chronology, named SP19, for the South Pole Ice Core (SPICECore), back to 54,302 yr BP. This chronology is based on a combination of 251 volcanic matching to the WAIS Divide ice core and annual layer counting (back to 11,341 yr BP for the latter). More precisely, the SP19 chronology is strictly tied to the

[Figure]

WD2014 chronology at the depth of the volcanic matches, and then layer counting is used to interpolate in-between. Before 11,341 yr BP, a spline interpolation method is used instead of annual layer counting. The layer counting is based on CFA measurements of magnesium, sodium, sulfate, chloride and dust. It has been performed by 4 different operators and reconciliation is found a posteriori when there is a discrepancy. A comparison is also made with visual stratigraphy but chemical stratigraphy is preferred because it is found to be more accurate.

It is argued that the WD2014 is used because it is more precise (annual layer thickness is larger at WD) and to have a WD2014 compatible time scale. The relative uncertainty to WD2014 is small during the Holocene, generally less than 18 yr and always less than 25 yr. For older parts, the relative uncertainty to WD2014 is less than 124 yr. The accumulation rate which is found in the SPICEcore is found to be mainly due to the upstream spatial pattern of accumulation along the flow line. The accumulation reconstruction is also compatible with nitrate concentration, nitrate amplitude of seasonal variations and N-15 of N2 in air bubbles (through a dynamical Herron-Langway firn model for the latter). It is therefore argued that the SPICEcore is a good candidate to test the influence of surface accumulation rate on the Lock-In Depth.

This article is very well written and its content is consistent. I therefore recommend to accept it.

–Thank you!

I only have a few technical corrections. l . 385: suppress "accumulation rate" since it is actually not plotted on this figure.

–Done.

l. 489, l. 531 and l. 535: double space after dot.

–Fixed.

---

## Author Comment (AC2) · 13 Aug 2019

Thank you very much for the very useful and positive comments. We greatly appreciate your input. Below are our line-by-line responses to your review.

In the paper "The SP19 Chronology for the South Pole Ice Core - Part 1: Volcanic matching and annual-layer counting" by D. A. Winsky and co-authors a new timescale (called SP19) for the SPICEcore is presented. This new time scale was partly achieved by annual layer counting but the main guide to build the age scale is a robust volcanic match coming from a comparison with WAIS Divide ice core chronology. Given the best quality of annual layering in WAIS Divide (as shown in figure 4) I think that the authors

[Figure]

choose the best methodological approach to build this time scale, considering WAIS as the most reliable annually counted scale. The discussion about the uncertainty of the age scale is really well done and confirms the goodness of this first SP19 age scale both for the Holocene period (with a maximum uncertainty of about 25 years) and for older ice (maximum 124 years in the longest time window without tie points). The paper is clear and well written and I recommend its publication after considering the following minor points.

–Thank you!

Minor comments: Figure 5: I would recommend to change the x-axis in kyr BP instead of using x10Ë4. In my opinion it would be much readable.

–We have changed the x-axis in Figure 5.

A list (table) of the volcanic horizons used to match WDC and SPICEcore would be valuable if inserted in the text or in supplementary material (even if archived at the NCDC or other repository).

–The timescale and a full list of tie points will be archived in the supplementary material and will be available at the National Climate Data Center (www.ncdc.noaa.gov) and the U.S. Antarctic Program Data Center (http://www.usap-dc.org). This data is also attached here.

Line 427: I would change this sentence to "Below 798 m depth (start of the Holocene): : :"

–Fixed.

Please also note the supplement to this comment:
https://www.clim-past-discuss.net/cp-2019-61/cp-2019-61-AC2-supplement.zip

---

## Author Comment (AC3) · 13 Aug 2019

Dear Anders Svensson,

Thank you very much for the very useful and positive comments. We greatly appreciate your input. Below are our line-by-line responses to your review.

Review by Anders Svensson of manuscript entitled 'The SP19 Chronology for the South Pole Ice Core – Part 1: Volcanic matching and annual-layer counting' submitted to Climate of the Past by D. Winsky et al.

The manuscript (MS) introduces a stratigraphic chronology SP19 of the South Pole

SPICE ice core based on 1) Holocene layer counting in high-resolution discrete chemistry samples and continuous records, and 2) a transfer of the WD2014 chronology based on identification of 251 common volcanic match points distributed over the last 54 ka. The layer counting is compared to a previously obtained independent layer counting from the same core based on visual stratigraphy alone. Furthermore, the authors are introducing an accumulation rate profile for the SPICE core based on a kink model. The model is compared to upstream accumulation patterns, to nitrate concentration profiles and to a d15N of N2 profile that all seem to support the obtained accumulation profile.

Overall, the MS is well written, well referenced and the figures are clear and illustrative of the study. The MS is well structured, the language is clear and the conclusions are well argued for.

–Thank you!

I only have a few comments below for the authors to consider. The authors perform multiple careful counting of annual layers of the Holocene using chemical parameters, continuous dust and conductivity. They then compare their resulting layer counting to an independent layer counting based on visual stratigraphy alone, and find an overall good agreement between the two approaches (Figure 9). Whereas this is a good test to see how well the two independent approaches are, it would probably have resulted in a better overall time scale, if all of the available high resolution records (chemistry + visual stratigraphy) had been combined in a common dating exercise from the beginning?

–We agree that it would have been informative to combine the visual and chemical layer counting from the beginning. However, the visual layer counting was completed two years prior to the chemistry layer counts and had already been tied independently to WD2014 with electrical conductivity before the chemistry data were available or before we had agreed on an overall dating strategy. Since the timescale was ultimately linked with WD2014 using sulfate and electrical conductivity, any minor changes to the

timescale resulting from an earlier reconciliation between methods would likely have little effect and would be within our uncertainty estimates.

Whereas I agree to the approach of transferring theWD2014 chronology to SPICE core rather than publishing a new independent time scale for SPICE core, it still seems like quite a large effort to do 4x independent layer counting of SPICE just to end up doing a transfer of time scale? Probably most of that time scale transfer could have been done based on a depth-depth matching alone (WDC – SPICE) similar to the approach taken in Figure 5?

–Yes, this was a lot of effort! We initially hoped to produce an independent timescale based on layer counting, but we ultimately decided to synchronize the timescale with WAIS Divide for the reasons described in section 3.1. However, we believe the effort was useful in assessing the uncertainty in the timescale.

There are certain depth intervals (228-275m and 626-687m), where all of the independent layer counting plus the automated Straticounter dating approach consistently count significantly fewer annual layers than suggested by the transfer of the WD2014 time scale. I understand that this consistent undercounting is associated with periods of exceptionally low accumulation (upstream) at SPICE. Are the authors convinced, however, that their layer counting is wrong and that there is not a problem with the WD2014 counting in one or both of those periods? In other words, can the authors account for all of the 'missing' layers when they go back and recount the critical sections in SPICE core? I'm not suggesting that the authors should revise the WD2014 time scale, but independent checks are always useful, and considering the effort put into precise dating of SPICE, an outcome may be suggestions for future revisions of the WDC chronology?

–This is a good question. We were able to account for all of the 'missing' layers needed during intervals where we had initially undercounted years. During these sections, the annual layering was less clear and many annual layers were missed by all five

interpreters. However, with the knowledge that there should be extra years in a given interval, it was never difficult to find examples of less well-expressed annual layers needed to synchronize the timescale. Furthermore, the missed layers often fell in intervals with smaller layer thickness (lower accumulation), making it difficult to resolve the annual cycle with the sampling frequency. The figure attached shows an example of such an area near 230 m, in a section that was initially very undercounted. This figure is equivalent to Figure 6 in the manuscript with sodium and magnesium plotted against the pick positions of the 5 interpreters. I have also added dotted lines to show where additional years were inserted for reconciliation. While many of the new years are poorly expressed in the chemistry or very thin, there is always some evidence present for each annual pick.

112,843 samples were collected and analyzed individually for this project! Did the authors consider doing fewer discrete and more continuous sample analysis? A CFA system optimized for depth/time resolution should be able to resolve the annual layering throughout the Holocene period. Of course, I'm not suggesting to do that now, but for future projects it may be an alternative?

–Thank you for the suggestion! It was, indeed a lot of work to process such a high volume of samples. Our decision to analyze discrete samples was based on previous positive experiences with discrete sampling, the expertise of the team and the availability of personnel. We believe our efforts were worthwhile due to the high quality of the resulting data.

A depth-difference relation figure between the two synchronized ice cores (SPICE, WDC) is very good for evaluating the synchronization and/or to identify regions where accumulation/thinning of the two records deviate. See Figure 2 of Seierstad et al., QSR, 2014: 'Consistently dated records from the Greenland GRIP, GISP2 and NGRIP ice cores for the past 104 ka reveal regional millennial-scale d18O gradients with possible Heinrich event imprint'.

–Thank you for this reference. Since we have elected to tie the SP19 record exactly to the WD2014 record, a precisely equivalent figure would show a depth differences dominated by the very different accumulation rates (∼7.4 cm/yr at SPICEcore and ∼20 cm/yr at WAIS Divide) as well as very different flow patterns (SPICEcore is hundreds of kilometers from a divide – WAIS Divide is much closer). For instance, in WAIS Divide, 10,000 BP is located at 1800 meters depth – deeper than the bottom of SPICEcore. However, Fig. S1 shows some similar information with the age offset at a given depth between the initial, unreconciled layer counts with WD2014. A similar diagram to Fig. S1, or configured as done by Seierstad et al. 2014, in our case, would show a nearly flat line with no offset since we forced the timescale within +/- 1 year of each tie point.

After having nicely synchronized the SPICE and WDC records, it would be nice to see the two climate profiles (water isotopes) in the same figure on their common time scale. If for whatever reason that is not possible, maybe the Calcium profiles of the two ice cores can be shown together? It is difficult for the reader to evaluate the quality of the volcanic matching without seeing a comparison of some parameter of the two ice core records.

–We have added this diagram to the supplemental material (Fig. S5) using calcium data.

Minor comments: Accumulation mistakenly used in figure 7 caption, already mentioned by Frederic.

–Fixed.

In Figure 3 is shown the seasonal variability of four impurities but not including nitrate. In section 4.5 and in Figure 11B the nitrate seasonality is discussed. Maybe it makes sense to include the seasonal variability of nitrate in Figure 3?

–We have added a 5th panel in Figure 3 to show the nitrate data.

In Figure 7, the annual layer thickness appears to stay constant or even increase

throughout the glacial part of the record. Wouldn't one normally expect a thinning of annual layering with depth?

–You are correct that one would expect layer thickness to decrease with depth for a constant accumulation rate. However, if the accumulation was higher for older ages, the extra thinning experienced by those will not necessarily offset the greater initial thickness. Thus, the layer thickness minimum near 25 ka BP could be due to accumulation rates that were substantially lower between 20-30 ka BP than they were beforehand. There is another possibility as well: the thinning function at the South Pole may be complex (i.e. not monotonic) due to the location hundreds of kilometers from a divide with irregular bedrock topography and converging/diverging flow. We have discussed Holocene accumulation rates here because they relate to our timescale accuracy and because they are much less sensitive to thinning. However, we have deliberately left a detailed interpretation of the layer thickness record and accumulation reconstruction for ice older than the Holocene to future studies (for instance Fudge et al. 2019 – CPD).

[Figure]

**Fig. 1.**